# Weak Scale Supersymmetry Emergent from the String Landscape

**DOI:** 10.3390/e26030275

**Published:** 2024-03-21

**Authors:** Howard Baer, Vernon Barger, Dakotah Martinez, Shadman Salam

**Affiliations:** 1Homer L. Dodge Department of Physics and Astronomy, University of Oklahoma, Norman, OK 73019, USA; dakotah.s.martinez-1@ou.edu; 2Department of Physics, University of Wisconsin, Madison, WI 53706, USA; barger@pheno.wisc.edu; 3Department of Mathematics and Natural Sciences, Brac University, Dhaka 1212, Bangladesh; ext.shadman.salam@bracu.ac.bd

**Keywords:** supersymmetry, string theory, landscape, lhc

## Abstract

Superstring flux compactifications can stabilize all moduli while leading to an enormous number of vacua solutions, each leading to different 4−d laws of physics. While the string landscape provides at present the only plausible explanation for the size of the cosmological constant, it may also predict the form of weak scale supersymmetry which is expected to emerge. Rather general arguments suggest a power-law draw to large soft terms, but these are subject to an anthropic selection of a not-too-large value for the weak scale. The combined selection allows one to compute relative probabilities for the emergence of supersymmetric models from the landscape. Models with weak scale naturalness appear most likely to emerge since they have the largest parameter space on the landscape. For finetuned models such as high-scale SUSY or split SUSY, the required weak scale finetuning shrinks their parameter space to tiny volumes, making them much less likely to appear compared to natural models. Probability distributions for sparticle and Higgs masses from natural models show a preference for Higgs mass $m_{h}∼125$ GeV, with sparticles typically beyond the present LHC limits, in accord with data. From these considerations, we briefly describe how natural SUSY is expected to be revealed at future LHC upgrades. This article is a contribution to the Special Edition of the journal *Entropy*, honoring Paul Frampton on his 80th birthday.

## 1. Introduction

Superstring theory provides the most promising avenue for unifying the Standard Model with gravity, but at the cost of requiring six or seven extra spatial dimensions [1,2,3,4,5]. The low energy limit $E<m_{P}$ (where $m_{P}$ is the reduced Planck mass) of string theory, once Kaluza–Klein modes are integrated out, is expected to be 10−d supergravity (SUGRA). The 10−d SUGRA theory is then assumed to be compactified to a tiny 6−d space *K* tensored with our usual 4−d (approximately) Minkowski spacetime $M_{4}$: $M_{10}=M_{4}\times K$. Originally, *K* was taken to be a 6−d compact Ricci-flat Kähler manifold with special holonomy [6]; such a Calabi–Yau manifold admits a conserved Killing spinor leading to a remnant N=1 supersymmetry (SUSY) on $M_{4}$.

The cosmological constant (CC) problem remained a thorny issue until the early 2000s when it was realized that string flux compactifications could lead to an enormous number of vacuum states each with different 4−d laws of physics, and in particular, different $\Lambda_{CC}$ values [7]. Such large numbers of vacuum states ($N_{vac}∼10^{500}$ is an oft-quoted number [8]) provided a setting for Weinberg’s anthropic solution to the CC problem [9]. But if the landscape [10] of string vacua provides a solution to the CC problem, might it also enter into other naturalness problems, such as the $m_{weak}/m_{P}$ (or related, $m_{SUSY}/m_{P}$) hierarchy problems (where $m_{P}≃2.4\times 10^{18}$ GeV)?

In this contribution to the volume of the journal *Entropy*, honoring Paul Frampton on his 80th birthday, we address this question. Here, we will put forward arguments in favor not only of weak scale SUSY as emergent from the string landscape, but indeed of a special form of weak scale SUSY (WSS)—SUSY with radiatively driven naturalness [11,12], or stringy-natural SUSY [13]. The specific form of WSS predicts at present that a light SUSY Higgs boson with mass $m_{h}≃125$ GeV should emerge, whilst sparticles masses are at present somewhat or well beyond the reach of the CERN Large Hadron Collider (LHC) [14]. It also allows us to predict a variety of SUSY signatures, which may allow for SUSY discovery at LHC luminosity upgrades over the coming years. Perhaps most important of these are the soft isolated opposite-sign dileptons+MET which arise from light higgsino pair production [15] and which recoil against a hard initial state jet radiation [16,17,18,19]. At present, both ATLAS [20] and CMS [21] with 139 ${\mathrm{fb}}^{-1}$ seem to have 2σ excesses in this channel, and an associated monojet signal may also be emerging [22].

## 2. Approximate Supersymmetric Vacua from String Theory

The main motivation for SUSY is that it provides a ’tHooft technical naturalness solution to the gauge hierarchy problem via the cancellation of quadratic divergences associated with the Higgs sector. This is true for SUSY breaking at any energy scale below $m_{P}$, since in the limit of $m_{SUSY}\to 0$, the model becomes more (super)symmetric. Thus, SUSY provides a technically natural solution to the so-called big hierarchy problem.

Specific motivation for weak scale SUSY comes from the little hierarchy problem and what we call *practical naturalness* [23,24]: an observable O is practically natural if all independent contributions to O are comparable to or less than O. For the case of WSS, we can relate the weak scale $m_{weak}∼m_{W,Z,h}∼100$ GeV to the weak scale soft SUSY-breaking terms and SUSY-conserving μ term via the scalar potential minimization conditions:(1)$$\frac{m_{Z}^{2}}{2}=\frac{m_{H_{d}}^{2}+\Sigma_{d}^{d}-\left(m_{H_{u}}^{2}+\Sigma_{u}^{u}\right)\tan^{2}\beta }{\tan^{2}\beta -1}-\mu^{2}∼-m_{H_{u}}^{2}-\Sigma_{u}^{u}\left({\tilde{t}}_{1,2}\right)-\mu^{2}$$
where $m_{H_{u,d}}^{2}$ are the soft SUSY-breaking Higgs masses and the $\Sigma_{u,d}^{u,d}$ contain an assortment of loop corrections to the scalar potential (explicit formulae are included in Ref’s [12]). A measure of practical naturalness $\Delta_{EW}$ can be defined which compares the largest (absolute) contribution to the right-hand side of Equation (Equation 1) to $m_{Z}^{2}/2$. Requiring $\Delta_{EW}≲30$ fulfills the practical naturalness condition. From Equation (Equation 1), we see immediately that $m_{H_{u}}^{2}$ must be driven to *small* negative values at the weak scale while the μ term must also be μ∼ 100–350 GeV. The latter condition means the higgsinos are usually the lightest SUSY particles, and the only ones required to be $∼m_{weak}$. The other sparticles enter via the $\Sigma_{u,d}^{u,d}$ terms, and hence, are suppressed by loop factors, and so can live in the TeV or beyond range. We shall see shortly that practical naturalness is closely linked to the selection of SUSY models on the landscape.

On the theory side, we expect the 4−d vacua emergent from the landscape to often contain some remnant SUSY.

*Remnant spacetime SUSY:* In Ref. [25], Acharya argues that all stable, Ricci-flat manifolds in dimensions <11 have special holonomy, and consequently a conserved Killing spinor. If so, then some remnant spacetime SUSY should exist in the 4−d low-energy effective field theory (LE-EFT).*EW stability:* A problem for the Standard Model to be the low-energy effective field theory for $E<m_{P}$ is electroweak stability, in that the Higgs quartic term λ may evolve to negative values at some intermediate scale, leading to a runaway scalar potential. For $m_{t}∼173.2$ GeV, the SM is just on the edge of metastability/runaway [26,27]. The Minimal Supersymmetric Standard Model (MSSM) has no such problem, since for the MSSM the quartic couplings involve the gauge couplings, which are always positive.*Landscape vacua stability:* In Ref’s [28,29], Dine et al. ask what sort of conditions can stabilize landscape deSitter vacua against decay to AdS vacua, leading to a big crunch. The presence of SUSY leads to absolutely stable vacua, whilst the presence of approximate (broken) SUSY leads to (metastable) vacua decay rates $\Gamma ∼m_{P}e^{-m_{P}^{2}/m_{3/2}^{2}}$ far beyond the age of the universe.*Hierarchy of scales:* While a hierarchy of scales is typically hard to come by in many BSM models, SUSY models allow for dynamical SUSY breaking [30], where the SUSY-breaking scale $m_{hidden}$ is gained via dimensional transmutation $m_{hidden}∼m_{P}e^{-8\pi^{2}/bg^{2}}$ and where the soft terms are developed as $m_{soft}∼m_{hidden}^{2}/m_{P}$ under gravity-mediation. Here, we only consider gravity-mediation, since gauge mediation leads to tiny trilinear soft terms *A*, which then require unnatural top-squark contributions $\Sigma_{u}^{u}$ to gain $m_{h}∼125$ GeV [31].*Harmony:* Witten emphasizes that consistent QFTs exist for spin-0, 1/2, 1, 3/2, and 2. The graviton is the physical spin-2 particle and the spin-3/2 Rarita–Schwinger gravitino field would exist as the superpartner of the graviton, thus filling out all allowed spin states.

In addition, WSS is motivated experimentally by a variety of measurements.

The measured values of the gauge couplings unify under MSSM RG evolution but do not under most other BSM extensions, including the SM itself [32].The measured top-quark mass is large enough to seed the required radiative breakdown of EW symmetry [33].The measured value of $m_{h}≃125$ GeV falls squarely into the range allowed by the MSSM: $m_{h}≲130$ GeV [34].Precision EW corrections tend to prefer the (heavy spectra) MSSM over the SM [35].

A complaint often made, with good reason, is that gravity mediation has its own flavor and CP problems, the former arising from operators such as $\int d^{4}\theta S^{†}SQ_{i}^{†}Q_{j}/m_{P}^{2}$, where *S* is a hidden sector superfield obtaining a SUSY-breaking vev $F_{S}∼10^{11}$ GeV and the $Q_{i}$ are visible-sector chiral superfields with generation index i,j=1−3. Since no symmetry forbids such flavor mixing, then FCNCs are expected to be large in gravity-mediated SUSY breaking (historically, this strongly motivated the search for flavor-conserving models such as gauge-mediation and sometimes anomaly-mediation). It is pointed out in Ref. [36] that the landscape provides its own decoupling/quasi-degeneracy solution to the SUSY flavor and CP problems by pulling first-/second-generation matter scalars to a flavor-independent upper bound in the 20–40 TeV range.

For these reasons, we will assume a so-called “fertile patch” or friendly neighborhood [37] of the string landscape: those vacua which include the MSSM as the LE-EFT and where only the CC and the magnitude of the weak scale scan within the landscape. In this case, Yukawa couplings and gauge couplings are instead fixed by string dynamics rather than environmental selection. This leads to predictive landscape models [37]: if the CC is too large, then large scale structure will not form, which seems required for complexity to emerge (the structure principle, leading to Weinberg’s successful prediction of $\Lambda_{CC}$). Only the magnitude of the weak scale scans. If $m_{weak}^{PU}≳4m_{weak}^{OU}$, then the down–up quark mass difference becomes so large that neutrons are no longer stable in nuclei and the only atoms formed in the early universe are hydrogen. If $m_{weak}^{PU}≲0.5m_{weak}^{OU}$, then we obtain a universe with only neutrons. This is the atomic principle [38], since complex nuclei are also apparently needed for complexity to emerge in any pocket universe (PU) within the greater multiverse (and where OU refers to $m_{weak}$ in our universe).

## 3. Natural SUSY from the Landscape

It is emphasized by Douglas that the CC scans independently of the SUSY-breaking scale in the landscape [39]. For the SUSY-breaking scale, we expect the vacua to be distributed as
(2)$$dN_{vac}∼f_{SUSY}\cdot f_{EWSB}\cdot dm_{SUSY}^{2}$$
where $m_{SUSY}$ is the overall hidden sector SUSY-breaking scale, expected to be $∼10^{11}$ GeV, such that the scale of soft terms is given by $m_{soft}∼m_{SUSY}^{2}/m_{P}$.

### 3.1. Distribution of Soft Breaking Terms on the Landscape

How is $f_{SUSY}$ distributed? Douglas [39] emphasizes that there is nothing in string theory to favor any particular SUSY-breaking vev over another, and hence, $m_{soft}$ would be distributed as a power-law:(3)$$f_{SUSY}∼m_{soft}^{2n_{F}+n_{D}-1}$$
where $n_{F}$ is the total number of (complex-valued) *F*-breaking fields and $n_{D}$ is the total number of (real-valued) *D*-breaking fields contributing to the overall scale of SUSY breaking, $m_{SUSY}^{4}=\sum_{i}{\left|F_{i}\right|}^{2}+\sum_{\alpha }D_{\alpha }^{2}$. The prefactor of 2 in the exponent arises because the $F_{i}$ are distributed randomly as complex numbers. For the textbook case of SUSY breaking via a single *F* term, then we expect $f_{SUSY}∼m_{soft}^{1}$, i.e., a *linear* draw to large soft terms. If more hidden fields contribute to the overall SUSY-breaking scale, then the draw to large soft terms will be a stronger power-law.

While the overall SUSY-breaking scale is distributed as a power-law, the different functional dependence [40,41,42] of the soft terms on the hidden sector SUSY-breaking fields means that gaugino masses, the trilinear soft terms, and the various scalar masses will effectively scan independently on the landscape [43]. Now, it is an *advantage* that different scalar mass-squared terms scan independently (as expected in SUGRA), since the first-/second-generation scalars are pulled to much higher values than third-generation scalars, while the two Higgs soft masses are also non-universal and scan independently. This situation is borne out in Nilles et al.’s mini-landscape, where different fields gain different soft masses due to their different geographical locations on the compactification manifold [44]. In terms of gravity mediation, then the so-called *n*-extra-parameter non-universal Higgs model (NUHMn) with parameters [45,46]
(4)$$m_{0}\left(i\right),\;m_{H_{u}},\;m_{H_{d}},\;m_{1/2},A_{0},\;\tan\beta \;\;\;\;\;\left(NUHM4\right)$$
provides the proper template. Since the matter scalars fill out a complete spinor rep of SO(10), we assume each generation i=1−3 is unified to $m_{0}\left(i\right)$. Also, for convenience one may ultimately trade $m_{H_{u}}$ and $m_{H_{d}}$ for the more convenient weak scale parameters $m_{A}$ and μ. One may also build (and scan separately) the natural anomaly-mediated SUSY-breaking model [47,48] (nAMSB) and the natural generalized mirage mediation model [49] (nGMM).

### 3.2. The ABDS Window

The anthropic selection on the landscape comes from $f_{EWSB}$. This involves a rather unheralded prediction of the MSSM: the value of the weak scale in terms of soft SUSY-breaking parameters and μ, as displayed in Equation (Equation 1). However, in the multiverse, here we rely on the existence of a friendly neighborhood [37], wherein the LE-EFT contains the MSSM but where only dimensionful quantities such as $\Lambda_{CC}$ and $v_{u}^{2}+v_{d}^{2}$ scan, whilst dimensionless quantities like gauge and Yukawa couplings are determined by dynamics. This assumption leads to *predictivity* as we shall see.

Under these assumptions, then we ask what conditions lead to complex nuclei, atoms as we know them, and hence, the ability to generate complex lifeforms in a pocket universe? For different values of soft terms, frequently one is pushed into a weak scale scalar potential with charge-or-color-breaking minima (CCB) where one or more charged or colored scalar mass squared is driven tachyonic (i.e., $m^{2}<0$). Such CCB minima must be vetoed. Also, for values of $m_{H_{u}}^{2}$ that are too large, then its value is *not* driven to negative values and EW symmetry is generally not broken. These we label as “no EWSB” and veto them as well. In practice, we must check the boundedness of the scalar potential from below in the vacuum stability conditions and that the origin of field space has been destabilized at least at tree level.

At this point, we are left with (MS)SM vacua where the EW symmetry is properly broken, but where $m_{weak}∼m_{W,Z,h}$ is at a different value from what we see in our universe. Here, we rely on the prescient analysis of Agrawal, Barr, Donoghue, and Seckel (ABDS) [38,50]. If the derived value of the weak scale is bigger than ours by a factor of 2–5, then the light quark mass difference $m_{d}-m_{u}$ becomes so large that neutrons are no longer stable in the nucleus and nuclei with Z≫N are not bound; such pocket universes would have nuclei of single protons only, and would be chemically inert. Likewise, if the PU value of the weak scale is a factor ∼0.5 less than our measured value, then one obtains a universe with only neutrons—also chemically inert. The ABDS window of allowed values is that
(5)$$0.5m_{weak}^{OU}<m_{weak}^{PU}≲4m_{weak}^{OU}$$
where we take the $\left(2-5\right)m_{weak}^{OU}$ to be $∼4m_{weak}^{OU}$ for definiteness, which is probably a conservative value. This is very central to our analysis and so is displayed in Figure 1. Our anthropic condition $f_{EWSB}$ is then that the scalar potential leads to minima with not only the appropriate EWSB, but also that the derived value of the weak scale lies within the ABDS window. Vacua not fulfilling these conditions must be vetoed. Early papers on this topic used instead a naturalness “penalty” of $f_{EWSB}∼m_{weak}^{2}/m_{SUSY}^{2}$; this condition would allow for many of the vacua which are forbidden by our approach.

### 3.3. EW-Natural SUSY Emergent from the Landscape

The next goal is to build a toy simulation of our friendly neighborhood of the string landscape. We can generate the soft terms of Equation (Equation 4) according to a power-law selection, usually taken to be $n=2n_{F}+n_{D}-1=1$ (linear draw). While Equation (Equation 3) favors the largest possible soft terms, the anthropic veto $f_{EWSB}$ places an upper bound on such terms because usually large soft terms lead to too large a value of $m_{weak}^{PU}$ beyond the ABDS window. The trick is to take the upper bound on scan limits beyond the upper bound posed by $f_{EWSB}$. However, in some cases larger soft terms are *more* apt to generate vacua within the ABDS window. A case in point is $m_{H_{u}}^{2}$: the smaller its value, the more negative it runs to unnatural values at the weak scale, while as it gets larger, then it barely runs negative: EW symmetry is barely broken. As its high scale value becomes even larger, it does not run negative by $m_{weak}$, and EW symmetry is typically not properly broken—such vacua failing to break the EW symmetry are vetoed. Also, for small $A_{0}$, the $\Sigma_{u}^{u}\left({\tilde{t}}_{1,2}\right)$ terms can be large. When $A_{0}$ becomes significantly negative, then cancellations occur in $\Sigma_{u}^{u}\left({\tilde{t}}_{1,2}\right)$ such that these loop corrections then lie within the ABDS window: large negative weak scale *A* terms make $\Sigma_{u}^{u}\left({\tilde{t}}_{1,2}\right)$ more natural while raising the light Higgs mass to $m_{h}∼125$ GeV.

A plot of the weak scale values of $m_{H_{u}}$ and μ is shown in Figure 2 (taken from Ref. [51]) for the case where all radiative corrections—some negative and some positive [52]—lie within the ABDS window. The ABDS window lies between the red and green curves. Imagine playing darts with this target, trying to land your dart within the ABDS window. There is a large region to the lower left where both $m_{H_{u}}$ and μ are ≲350 GeV, which leads to PUs with complexity. Alternatively, if you want to land your dart at a point with μ∼1000 GeV, then the target space has pinched off to a tiny volume: the target space is finetuned and your dart will almost never land there. The EW-natural SUSY models live in the lower-left ABDS window while finetuned SUSY models with large $\Delta_{EW}$ lie within the extremely small volume between the red and green curves in the upper-right plane. This tightly constrained region is labeled by split SUSY [53], high-scale SUSY [54] and mini-split [55].

It is often said that landscape selection offers an alternative to naturalness and allows for finetuned SUSY models. After all, is the CC not finetuned? However, from Figure 2 we see that models with EW naturalness (low $\Delta_{EW}$) have a far greater relative probability to emerge from landscape selection than finetuned SUSY models.

In Figure 3 (from Ref. [51]), we perform a numerical exercise to generate high-scale SUSY soft terms in accord with an n=1 draw in Equation (Equation 3). The green dots are viable vacua states with appropriate EWSB and $m_{weak}^{PU}$ within the ABDS window. While some dots land in the finetuned region, the bulk of the points lie within the EW-natural SUSY parameter space.

An alternative view is gained from Figure 4 from Ref. [56]. Here, we compute contributions to the scalar potential within a variety of SUSY models including RNS (radiatively driven natural SUSY [11]), CMSSM [57], G_2_MSSM [58], high-scale SUSY [59], spread SUSY [60], mini-split [55], split SUSY [53], and the SM with cutoff $\Lambda =10^{13}$ TeV, indicative of the neutrino see-saw scale [61]. The *x*-axis is either the SM μ parameter or the SUSY μ parameter while the *y*-axis is the calculated value of $m_{Z}$ within the PU. The ABDS window is the horizontal blue-shaded region. For μ distributed as equally likely at all scales (the distribution’s probability density integrates to a log), then the length of the *x*-axis interval leading to $m_{Z}^{PU}$ within the ABDS window can be regarded as a relative probability measure $P_{\mu }$ for the model to emerge from the landscape. There is a substantial interval for the RNS model, but for finetuned SUSY models, the interval is typically much more narrow than the width of the printed curves. We can see that finetuned models have only a tiny range of μ values which allow habitation within the ABDS window.

Using the magic of algebra, the width of the μ intervals can be computed, and the results are shown in Table 1. Here, $P_{\mu }$ is to be considered as a *relative* probability. From the table, we see that the SM is about $10^{-27}$ times less likely to emerge as compared to RNS. Mini-split is $10^{-4}$–$10^{-8}$ times less likely to emerge (depending on the version of mini-split). Even the once-popular CMSSM model is $∼10^{-3}$ times less likely than RNS to emerge from the landscape.

## 4. Radiatively Driven Natural SUSY

Along with probability distributions for models to emerge from the landscape, one can compute probability distributions for sparticle and Higgs mass values from particular models given an assumed value of *n* in $f_{SUSY}$. Here, we use a linear draw, n=1, to large soft terms with the NUHM4 model as the LE-EFT. We capture non-finetuned models by requiring $\Delta_{EW}≲30$, i.e., that the largest independent contribution to $m_{Z}$ lies within the ABDS window. These models have radiatively driven naturalness (RNS) [11], where RG running drives various soft terms to natural values at the weak scale. The value of $m_{H_{u}}\left(m_{GUT}\right)$ is selected so that $m_{Z}=91.2$ GeV in our universe.

The distribution for the light Higgs mass is shown in Figure 5. We see for n=1 that the blue distribution rises to a maximum at $m_{h}∼125$ GeV. This is where $A_{t}$ is large enough to yield cancellations in the $\Sigma_{u}^{u}\left({\tilde{t}}_{1,2}\right)$ terms, but also lifts $m_{h}$ up to ∼125 GeV via maximal stop mixing [11]. For comparison, we also show the orange histogram for n=−1, where soft terms are equally favored at any mass scale. Here, the distribution peaks at $m_{h}∼118$ GeV, with hardly any probability at $m_{h}∼125$ GeV.

In Figure 6, we show the corresponding probability distribution for the gluino mass. Here, for n=1 the curve begins around $m_{\tilde{g}}∼1$ TeV and reaches a broad maximum around 3–4 TeV, while petering out beyond $m_{\tilde{g}}∼6$ TeV. The present LHC Run 2 limit from ATLAS/CMS [62,63] is $m_{\tilde{g}}≳2.2$ TeV from searches within the simplified model context. From the plot, we see that the LHC is only beginning to probe the expected range of $m_{\tilde{g}}$ values from the landscape.

Other sparticle and Higgs mass distributions from the landscape are shown in Refs. [14,64], and they are typically beyond or even well beyond present LHC limits. For instance, light top-squarks are expected around $m_{{\tilde{t}}_{1}}$∼1–2.5 TeV whilst first-/second-generation squarks and sleptons are expected near $m_{\tilde{q},\tilde{\ell }}$∼10–30 TeV. From this point of view, LHC is at present seeing exactly what the string landscape predicts.

## 5. Conclusions

Theoretical arguments suggest many models which include a remnant spacetime SUSY to populate the string landscape of 4−d vacua. We assume a friendly neighborhood of the landscape populated by the MSSM as the LE-EFT, but where the CC and also the soft SUSY-breaking gaugino masses, scalar masses, and *A*-terms scan via a power-law draw to large values. Landscape selection of soft terms then allows for a derived value of the weak scale, which must lie within the ABDS window in order for the atomic principle to be obeyed, leading to complex nuclei, and hence, atoms which are needed for complexity.

Under the landscape selection of soft SUSY-breaking terms, one expects radiative natural SUSY, or RNS, to be much more prevalent than finetuned SUSY models such as CMSSM, G_2_MSSM, high-scale SUSY, split SUSY, or mini-split SUSY. This is evident because in RNS, where all contributions to the weak scale lie within the ABDS window, there is a much larger volume of scan space leading to $m_{weak}\in ABDS$. Alternatively, if even one contribution to the weak scale lies outside the ABDS window, then the remaining volume of parameter space leading to $m_{weak}\in ABDS$ shrinks to tiny values, and is relatively less likely. This is borne out by toy simulations of the string landscape and also allows for a relative probability measure $P_{\mu }$ for different models to emerge from the landscape. For instance, $P_{\mu }\left(RNS\right)∼1.4$, compared to, for instance, $P_{\mu }\left(HS-SUSY\right)∼6\times 10^{-4}$. Finally, we show probability distributions of the light Higgs mass and gluinos, showing that the present LHC is seeing what one would expect from the string landscape. New SUSY signals, especially from higgsino pair production, could arise within the next few years at LHC. With all these beautiful results, we anticipate that Paul will begin to work on landscape SUSY as well.

## Figures and Tables

**Figure 1 entropy-26-00275-f001:**
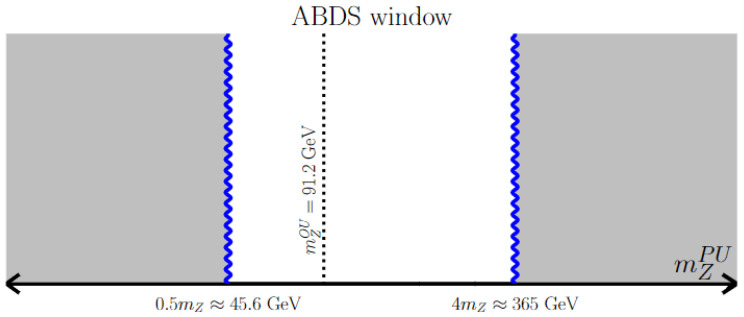
The ABDS-allowed window within the range of $m_{Z}^{PU}$ values.

**Figure 2 entropy-26-00275-f002:**
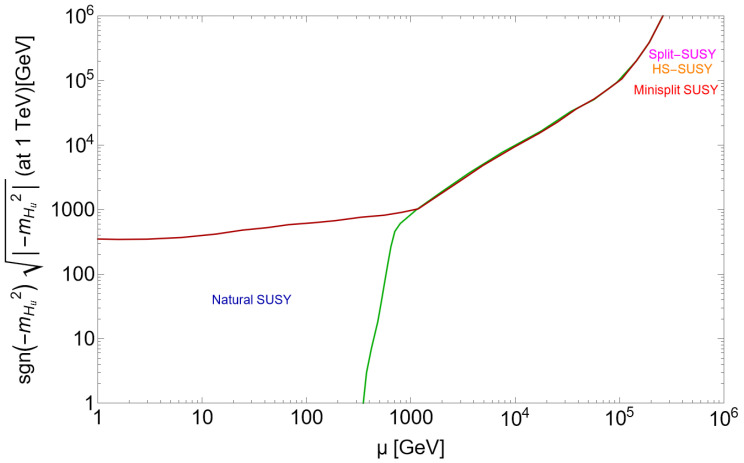
The $\mu^{PU}$ vs. $\sqrt{-m_{H_{u}}^{2}\left(weak\right)}$ parameter space in a toy model ignoring radiative corrections to the Higgs potential. The region between the red and green curves leads to $m_{weak}^{PU}<4m_{weak}^{OU}$ so that the atomic principle is satisfied.

**Figure 3 entropy-26-00275-f003:**
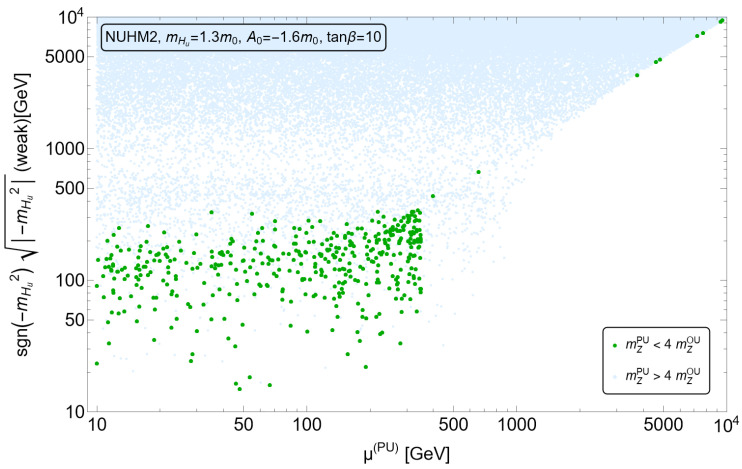
The value of $m_{H_{u}}\left(weak\right)$ vs. $\mu^{PU}$ The green points denote vacua with appropriate EWSB and with $m_{weak}^{PU}<4m_{weak}^{OU}$ so that the atomic principle is satisfied. Blue points have $m_{weak}^{PU}>4m_{weak}^{OU}$.

**Figure 4 entropy-26-00275-f004:**
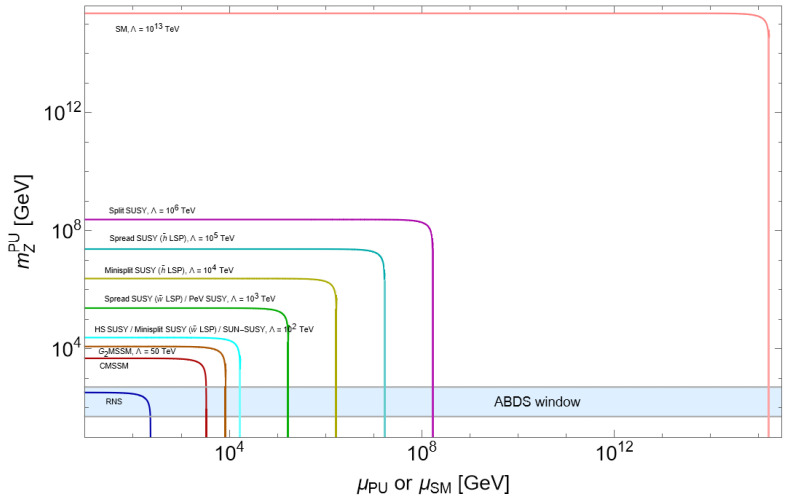
Values of $m_{Z}^{PU}$ vs. $\mu_{PU}$ or $\mu_{SM}$ for various natural (RNS) and unnatural SUSY models and the SM. The ABDS window extends here from $m_{Z}^{PU}∼50$ to 500 GeV.

**Figure 5 entropy-26-00275-f005:**
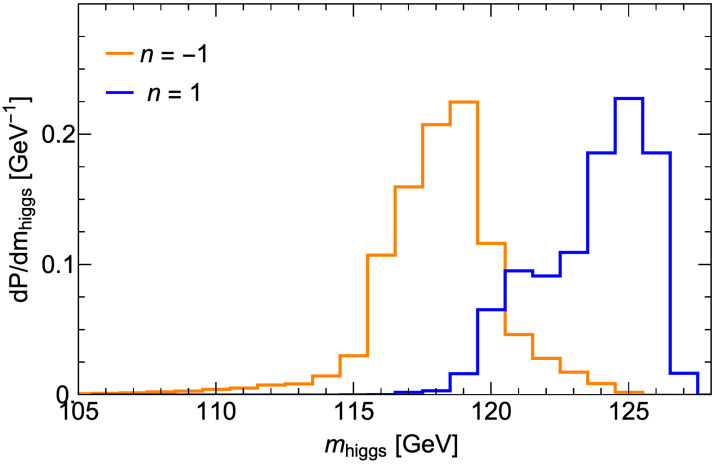
Probability distributions for the light Higgs scalar mass $m_{h}$ from the $f_{SUSY}=m_{soft}^{\pm 1}$ distributions of soft terms in the string landscape with μ=150 GeV.

**Figure 6 entropy-26-00275-f006:**
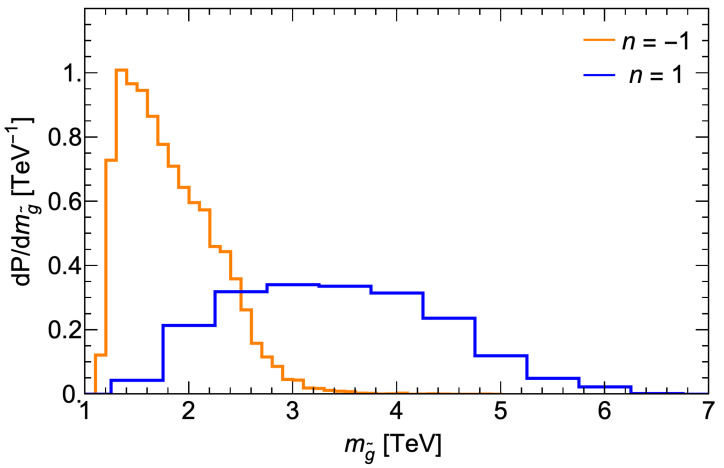
Probability distribution for $m_{\tilde{g}}$ from the $f_{SUSY}=m_{soft}^{\pm 1}$ distributions of soft terms in the string landscape with μ=150 GeV.

**Table 1 entropy-26-00275-t001:** A survey of some unnatural and natural SUSY models along with general expectations for sparticle and Higgs mass spectra in TeV units. We also show the relative probability measure $P_{\mu }$ for the model to emerge from the landscape. For RNS, we take $\mu_{min}=10$ GeV.

Model	$\tilde{m}\left(1,2\right)$	$\tilde{m}\left(3\right)$	Gauginos	Higgsinos	$m_{h}$	$P_{\mu }$
SM	-	-	-	-	-	$7\times 10^{-27}$
CMSSM ($\Delta_{EW}=2641$)	∼1	∼1	∼1	∼1	0.1–0.13	$5\times 10^{-3}$
PeV SUSY	∼$10^{3}$	∼$10^{3}$	∼1	$1-10^{3}$	0.125–0.155	$5\times 10^{-6}$
Split SUSY	∼$10^{6}$	∼$10^{6}$	∼1	∼1	0.13–0.155	$7\times 10^{-12}$
HS-SUSY	≳$10^{2}$	≳$10^{2}$	≳$10^{2}$	≳$10^{2}$	0.125–0.16	$6\times 10^{-4}$
Spread ($\tilde{h}$LSP)	$10^{5}$	$10^{5}$	$10^{2}$	∼1	0.125–0.15	$9\times 10^{-10}$
Spread ($\tilde{w}$LSP)	$10^{3}$	$10^{3}$	∼1	∼$10^{2}$	0.125–0.14	$5\times 10^{-6}$
Mini-Split ($\tilde{h}$LSP)	∼$10^{4}$	∼$10^{4}$	∼$10^{2}$	∼1	0.125–0.14	$8\times 10^{-8}$
Mini-Split ($\tilde{w}$LSP)	∼$10^{2}$	∼$10^{2}$	∼1	∼$10^{2}$	0.11–0.13	$4\times 10^{-4}$
SUN-SUSY	∼$10^{2}$	∼$10^{2}$	∼1	∼$10^{2}$	0.125	$4\times 10^{-4}$
G_2_MSSM	30–100	30–100	∼1	∼1	0.11–0.13	$2\times 10^{-3}$
RNS/landscape	5–40	0.5–3	∼1	0.1–0.35	0.123–0.126	1.4

## Data Availability

No new data were created or analyzed in this study. Data sharing is not applicable to this article.

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
