# Peer review of "Weak Scale Supersymmetry Emergent from the String Landscape"

_entropy, 2024, doi:10.3390/e26030275_

Round 1

Reviewer 1 Report

Comments and Suggestions for Authors

This work discusses the consequences for the supersymmetric standard model from combining various anthropic and string landscape arguments.  The arguments are mostly summarized, but two points are discussed in more detail.  The first is an anthropic condition on the weak scale due to Agrawal, Barr, Donoghue and Seckel which requires it to be near the observed value (more precisely in a window [0.5,5] times that value).  Following the authors' [51], it is argued that this condition much favors natural low energy susy over fine tuned scenarios.  The second is the power law distribution of supersymmetry breaking soft terms in the landscape.  The expected value of the Higgs mass depends strongly on these, and the observed value favors a linear measure for the soft breaking terms.  This is interesting as the usual assumption is a log measure. 

The overall conclusion, that the combination of anthropic and landscape arguments favors natural susy, is rather surprising and important if true.  For this reviewer, there are too many assumptions (for example that the mu parameter is log distributed) to make it convincing, but it is stimulating.

In summary, a nice introduction to a rather intricate set of arguments with potentially important consequences, which hopefully will bring them to a wide audience.

Reviewer 2 Report

Comments and Suggestions for Authors

The submission is a good review of the current status of supersymmetry breaking in connection to superstring landscape. The authors advocate the weak scale supersymmetry breaking by using the naturalness arguments. The novel results are devoted to the radiatively-driven natural supersymmetry that is apparently consistent with the known Higgs mass. The expected consequences of the proposed scenario for the future searches of supersymmetric particles at LHC are presented.